# Loss of microRNA-135b Enhances Bone Metastasis in Prostate Cancer and Predicts Aggressiveness in Human Prostate Samples

**DOI:** 10.3390/cancers13246202

**Published:** 2021-12-09

**Authors:** Mireia Olivan, Marta Garcia, Leticia Suárez, Marc Guiu, Laura Gros, Olga Méndez, Marina Rigau, Jaume Reventós, Miguel F. Segura, Inés de Torres, Jacques Planas, Xavier de la Cruz, Roger R. Gomis, Juan Morote, Ruth Rodríguez-Barrueco, Anna Santamaria

**Affiliations:** 1Translational Oncology Laboratory, Anatomy Unit, Department of Pathology and Experimental Therapy, School of Medicine, Universitat de Barcelona (UB), 08907 L’Hospitalet de Llobregat, Spain; rrodriguezb@ub.edu; 2Molecular Mechanisms and Experimental Therapy in Oncology-Oncobell Program, Bellvitge Biomedical Research Institute (IDIBELL), 08908 L’Hospitalet de Llobregat, Spain; 3Cell Cycle and Cancer Laboratory, Biomedical Research Group in Urology, Vall d’Hebron Research Institute (VHIR), Universitat Autònoma de Barcelona (UAB), 08193 Bellaterra, Spain; marta.garcial@sjd.es (M.G.); leticia.suarez@vhir.org (L.S.); laura.gros@mafrica.es (L.G.); olga.mendez@vhir.org (O.M.); itorres@vhebron.net (I.d.T.); jplanas@vhebron.net (J.P.); jmorote@vhebron.net (J.M.); 4Developmental Tumor Biology Laboratory, Institut de Recerca Sant Joan de Déu, 08950 Esplugues de Llobregat, Spain; 5Cancer Science Programme, Institute for Research in Biomedicine (IRB-Barcelona), 08028 Barcelona, Spain; marc.guiu@irbbarcelona.org (M.G.); roger.gomis@irbbarcelona.org (R.R.G.); 6Bellvitge Biomedical Research Institute (IDIBELL), 08908 L’Hospitalet de Llobregat, Spain; mrigau@idibell.cat (M.R.); jreventos@uic.es (J.R.); 7Centro de Investigación Biomédica en Red de Cáncer (CIBERONC), 28029 Madrid, Spain; 8Departament de Ciències Bàsiques, Universitat Internacional de Catalunya, 08017 Barcelona, Spain; 9Group of Translational Research in Child and Adolescent Cancer, Vall d’Hebron Research Institute (VHIR), Universitat Autònoma de Barcelona (UAB), 08035 Barcelona, Spain; miguel.segura@vhir.org; 10Department of Pathology, University Hospital Vall d’Hebron, Universitat Autònoma de Barcelona (UAB), 08035 Barcelona, Spain; 11Department of Urology, University Hospital Vall d’Hebron, Universitat Autònoma de Barcelona (UAB), 08035 Barcelona, Spain; 12Institució Catalana de Recerca i Estudis Avançats (ICREA), 08010 Barcelona, Spain; xavier.delacruz@vhir.org; 13Group of Clinical and Translational Bioinformatics, Vall d’Hebron Research Institute (VHIR), Universitat Autònoma de Barcelona (UAB), 08035 Barcelona, Spain

**Keywords:** miRNA-135b, prostate cancer, bone metastasis, miRNAs

## Abstract

**Simple Summary:**

Prostate cancer (PCa) is the most prevalent cancer in males worldwide, and it was the fifth leading cause of cancer mortality in this group in 2020. Near 70% of advanced-stage PCa patients will undergo bone metastasis, suffering pathological complications that severely affect patients’ quality of life and probably progress in most cases to lethal PCa. Our main objective was to unveil novel molecules associated with choosing the bone as a metastatic niche. For this purpose, we generated and characterized a cell line with increased tropism to bone. Its molecular analysis has led us to identify factors with a potential role in bone metastasis that could also be used as biomarkers of disease progression. These data help us to understand the mechanisms that increase bone metastasis penetrance of PCa cells and could provide new therapeutic tools in the future for patients with worse prognoses.

**Abstract:**

About 70% of advanced-stage prostate cancer (PCa) patients will experience bone metastasis, which severely affects patients’ quality of life and progresses to lethal PCa in most cases. Hence, understanding the molecular heterogeneity of PCa cell populations and the signaling pathways associated with bone tropism is crucial. For this purpose, we generated an animal model with high penetrance to metastasize to bone using an intracardiac percutaneous injection of PC3 cells to identify PCa metastasis-promoting factors. Using genomic high-throughput analysis we identified a miRNA signature involved in bone metastasis that also presents potential as a biomarker of PCa progression in human samples. In particular, the downregulation of miR-135b favored the incidence of bone metastases by significantly increasing PCa cells’ migratory capacity. Moreover, the *PLAG1*, *JAKMIP2*, *PDGFA*, and *VTI1b* target genes were identified as potential mediators of miR-135b’s role in the dissemination to bone. In this study, we provide a genomic signature involved in PCa bone growth, contributing to a better understanding of the mechanisms responsible for this process. In the future, our results could ultimately translate into promising new therapeutic targets for the treatment of lethal PCa.

## 1. Introduction

Bone metastasis is a common complication of cancer and occurs in around 70% of patients with metastatic prostate cancer (PCa), PCa being the second-most commonly diagnosed cancer in men in the western world [1]. Generally, patients with advanced-stage PCa are treated with androgen deprivation therapy (ADT). Unfortunately, ADT is only temporarily effective in most cases; PCa gradually acquires resistance to androgen deprivation and eventually becomes castration-resistant PCa (CRPC), which is often associated with tumor dissemination [2]. This type of cancer is frequently multifocal, and it is associated with a higher grade, stage, and recurrence rate [3]. Such heterogeneity has an impact on PCa progression since individual cancer cells contained within the tumor may have distinct genomic profiles and, consequently, influence the efficacy of current and future therapies [3,4].

Bone metastasis results in devastating consequences for patients such as pain, hypercalcemia, and spinal cord and nerve-compression syndromes; importantly, after metastasis, tumors become generally incurable [5]. Each metastatic microenvironment favors or opposes colonization by disseminated tumor cells. Therefore, assuming that disseminated tumor cells that populate the bone marrow arise from primary tumors and could be found in early stages of the disease, understanding the distinct organ-specific mechanisms that enable metastatic growth is paramount. For this, clonal selection strategies based on preclinical models that faithfully reproduce some stages of bone metastasis have been developed [6,7]. These models provide us with useful tools to better understand the molecular basis underlying the orchestration of bone metastasis from initiation to development of distant dissemination and lead us to engineer solutions to improve outcomes for PCa patients.

Small non-coding microRNAs (miRNAs) are master regulators of gene expression which have the capacity to regulate multiple genes simultaneously and, thus, to redirect biological pathways [8]. Due to their properties, miRNAs emerged as promising tools for diagnosis, prognosis determination, and management of cancer patients [9,10]. In recent years, numerous studies have identified that the dysregulation of miRNA expression in cancer cells can modulate multiple aspects of cancer progression [11,12,13,14,15], including the bone metastasis process [8,16,17]. To date, some miRNAs have already been identified as metastasis suppressors in PCa, including miR-33a [18], the miR-34 family [19], miR-145 [20], miR-212 [21], and the miR-200 family [22]. On the other hand, a set of miRNAs has been identified as metastasis-promoting miRNAs such as miR-9 [23], miR-181a [24], miR-210-3 [25], and miR-454 [26]. Zhu et al. identified the miR-636 that is up-regulated in bone metastatic PCa tissue, and they found that this miRNA could promote migration and invasion by targeting MBNL2, TNS1, and STAB1 [17,27]. Of particular relevance to this study, miR-135b has been reported to be aberrantly expressed and to play different roles in the progression of a variety of cancers in a context-dependent manner. It has been shown that miR-135b has an oncogenic role in non-small-cell lung cancer [28], gastric cancer [29], colon cancer [30], and pancreatic cancer [31], among other tumor types. Contrary to this, miR-135b has also been identified as a tumor-suppressor miRNA in breast cancer [32] and osteosarcoma [33]. Focusing on the PCa context, only a few studies have been conducted to rule out the function of miR-135b on androgen receptor (AR) regulation [34,35,36]. Interestingly, Wang et al. [37] found that the expression of miR-135b was associated with tumor metastasis suppression by targeting STAT6. In summary, despite the progress of miRNAs that have been incorporated as diagnostic and therapeutic targets in clinical trials for solid tumors [38,39], to date none of them are aimed at targeting PCa metastasis [40]. Therefore, it is important to shed light on which miRNAs regulate tumorigenesis and metastasis, and how those miRNAs do this, so that they can be included in the future management of patients with PCa metastasis.

In this study, we report the establishment and characterization of a high-penetrant PCa bone-metastasizing cell line after several rounds of intracardiac injection in mice. In vivo selection gave rise to aggressive bone-seeking variants of PCa cells with an increased ability to grow in the bone. Using genomic high-throughput expression analysis and functional studies, we found evidence for a role of miR-135b downregulation in highly metastatic tumor cells suggesting that populations with low levels of miR-135b present in the primary tumors could potentiate the bone metastatic spread. Restoration of miR-135b in PCa cells decreased their migratory capacity. Furthermore, we found *PLAG1*, *JAKMIP2*, *PDGFA*, and *VTI1b* to be directly regulated by miR-135b and to be potential downstream effectors of miR-135b phenotypic effects. In summary, our data unveil a novel key regulator of bone metastatic PCa dissemination and enable future characterization of the dynamic changes in the bone metastatic niche, suggesting new therapeutic targets to be considered for the treatment of this lethal disease.

## 2. Materials and Methods

### 2.1. Cell Lines

The human PCa cell line PC3 was obtained from the American Type Culture Collection (#CRL-1435, ATCC, Rockville, MD, USA). The PC3 bone-metastatic clone (PC3-BM) was isolated from bone metastases produced in nude mice after intracardiac injection of PC3 cells. Both parental PC3 (PC3-P) and clone-derivative cells carried the firefly luciferase gene coding region cloned upstream of the green fluorescent protein (GFP) gene as described before [41]. PCa cells were cultured at 37 °C in a humidified atmosphere (5% CO_2_/95% air), in complete media RPMI-1640 (Thermo Fisher Scientific, Waltham, MA, USA) supplemented with 10% fetal bovine serum (FBS), 50 U/mL penicillin and streptomycin, 1X non-essential amino acids (MEM), 2 mM l-glutamine, 10 mM HEPES (all from Gibco), and 1 mM sodium pyruvate (PAA Laboratories, Manchester, UK).

The osteoblast cell line (hFOB 1.19) was kindly provided by Dr. Javier Garcia Castro (Cellular Biotechnology Unit from the Instituto de Salud Carlos III) and cultured with 1:1 mixture of Ham’s F12 Medium Dulbecco’s Modified Eagle’s Medium, with 2.5 mM l-glutamine (without phenol red) supplemented with 1% G418 (Thermo Fisher Scientific, Waltham, MA, USA) and fetal bovine serum to a final concentration of 10%.

### 2.2. Animal Model

Congenitally athymic male nude mice (Hsd.Athymic Nude-*Foxn1^nu^*) were purchased from Harlan Laboratories (Italy) at 4 weeks of age and maintained under specific pathogen-free conditions. The animals were kept in a sterile environment in cages with beds of sterilized soft wood granulate and fed an irradiated rodent diet *ad libitum* with autoclaved tap water. All the procedures associated with experimentation and animal care were performed according to the guidelines of the Spanish Council for Animal Care and the protocols of the ethics committee for animal experimentation at our own institution.

Five-week-old male mice were deeply anesthetized with ketamine (100 mg/kg of body weight) and xylazine (10 mg/kg of body weight) solution, and PC3 luciferase-transfected cells (3 × 10^5^ in 0.1 mL sterile PBS) were inoculated by intracardiac (i.c.) injection using a 25-gauge syringe. The presence of a rapid pulsatile flow of bright red arterial blood into the syringe was indicative of correct needle placement, as opposed to darker, burgundy-colored blood [42]. Mice were imaged for luciferase activity immediately after injection and continued to be monitored weekly using IVIS^®^ imaging. Animals were sacrificed by cervical dislocation 6 weeks post-injection, or earlier if there were early signs of serious distress. As a quality control, photon flux for each mouse was measured by the number of highlighted pixels calculated by using a circular region of interest (ROI) in a ventral or dorsal position and normalized to the value obtained immediately after xenografting at the same area (day 0) of each mouse, so that all mice had an arbitrary starting bioluminescence signal of 1. Bioluminescence imaging (BLI) was assessed to monitor tumor burden and tumor cell localization. Tumor-induced bone destruction was analyzed by micro-CT imaging using a Skyscan 1076 desktop microtomograph with high resolution capture and 3D modelling (Skyscan, Kontich, Belgium). After the mice were sacrificed, tumor cells from bone metastases were localized by ex vivo BLI imaging and freshly harvested under sterile conditions using a flushing procedure. Bone marrow and tumor cells were rinsed with PBS and put in cell culture dishes with geneticin selection. Attached cells were expanded in vitro and selected by GFP fluorescence-activated cell sorting using the FacsAria (BD Bioscience, San Jose, CA, USA) and reinjected via i.c. injection into a second set of animals. This process, based on the in vivo selection of metastatic PC3 cells, was repeated three times (*n* = 5 mice/round).

### 2.3. Study Subjects

This study was approved by the Clinical Research Ethics Committee from Vall Hebron University Hospital (PR(AG)96-2015). Written informed consent was obtained from all the study participants and samples were coded to ensure sample tracking and confidentiality of patient/donor identity. A total of 42 formalin-fixed and paraffin-embedded (FFPE) samples from radical prostatectomies (RP) were available from the archives of the Pathology Department of the Vall Hebron University Hospital (Barcelona, Spain) and kindly provided by Antonio López-Guerrero and Jose Rubio-Briones from Instituto Valenciano de Oncología (Valencia, Spain); the clinic–pathologic features of these patients are detailed in Appendix A. FFPE samples from patients who underwent RP were divided into two study groups: (i) patients without biochemical recurrence (BCR) after ≥10 years of follow up (No-BCR) and (ii) patients who had BCR (1–5 years after RP). BCR is defined as the first post-operative PSA of >0.4 ng/mL, as confirmed by at least one subsequent increasing value (persistent PSA increase) after achieving undetectable PSA post-operatively.

Plasma samples were obtained from 40 men with suspicion of PCa according to abnormal digital rectal examination (DRE) and/or serum PSA levels higher than 4 ng/mL, referred for a first prostate biopsy at the Urology Department of the Vall Hebron University Hospital (Barcelona, Spain). Patients with a diagnosis of PCa were divided into three groups: (i) localized PCa, (ii) locally advanced PCa, and (iii) metastatic PCa. Moreover, age-matched controls were selected. The clinic–pathologic features of these patients are detailed in Appendix A.

#### 2.3.1. Laser Capture Microdissection (LCM)

A slide of each FFPE sample was stained with hematoxylin, and an experienced pathologist selected the area of the tumor that presented the most predominant pattern of Gleason Score. To ensure that the entire sample used for subsequent RNA extraction was only tumoral tissue, laser capture microdissection was performed. Areas with high density and purity of tumor cells (>80%) were marked for microdissection of adjacent sections to minimize contamination from surrounding healthy prostate tissue. Overlapping areas in up to six adjacent slides were microdissected using a Leica LMD 6000 microdissection system (Leica Microsistemas S.L.U, L’Hospitalet de Llobregat, Spain). Samples were obtained from 10 to 15 µm unstained histological sections under the guidance of a hematoxylin-stained slide using an FSX100 System Olympus microscope (Olympus, Shinjuku City, Tokyo) and placed into sterile tubes for a subsequent RNA extraction.

#### 2.3.2. Plasma Collection

Blood samples were collected in EDTA-containing tubes at room temperature and processed for plasma isolation within 2 h. Blood samples were centrifuged at 4 °C for 20 min at 1500× *g*. Plasma was then aliquoted and stored at −80 °C within 24 h.

### 2.4. In Vitro Assays

#### 2.4.1. Transient Transfection of miRNA Mimics

Cells were transfected using synthetic oligonucleotides miRIDIAN Mimic (Dharmacon; Thermo Fisher Scientific, Waltham, MA, USA) with a final concentration of 25 nM. Transfection was performed using Lipofectamine 2000 (Invitrogen, Life Technologies, Waltham, MA, USA) following the manufacturer’s instructions. To obtain transfection vesicles, the lipofectamine and oligonucleotides were diluted and mixed in an OPTI-MEM medium (Gibco) and incubated for 20 min, then the mix was added to cell cultures and incubated for 12h. Following that, the medium was changed to avoid lipofectamine toxicity.

#### 2.4.2. Proliferation Analysis

For the proliferation experiments, 96 h post-transfection cells were seeded at 5 × 10^3^ cells per well on a 96-well plate (*n* = 6/condition). At the indicated time point, cells were fixed in a 4% formaldehyde solution and stained with 0.5% crystal violet. Crystals were dissolved with 15% acetic acid and the optical density was read at 590 nm using the Epoch Microplate Spectrophotometer (Agilent Technologies, Santa Clara, CA, USA).

#### 2.4.3. Transwell Migration Assay

For co-culture experiments, osteoblasts were seeded (5 × 10^4^/well) at the bottom of a 24-well plate with complete media for 24 h. The following day, PCa cells were seeded (1.0 × 10^5^ cells/ insert) onto transwell inserts (BD Bioscience, San Jose, CA, USA) with 8 μm diameter pore membranes in a medium supplemented with 2% FBS in the presence or absence of osteoblast-containing wells. Cells were allowed to migrate for 20 h at 37 °C and then fixed with 4% formaldehyde for 30 min at room temperature (RT). Cells that had attached to the membrane but not migrated were completely removed using a cotton swab. Migrated cells were stained with 1 µL/mL Hoechst 33258 (Sigma-Aldrich, St Louis, MO, USA) for 10 min at RT and counted.

#### 2.4.4. 3′-UTR Cloning and Luciferase Assay

The 3′-UTR regions of *VTI1b*, *PLAG1*, *JAKMIP2*, and *PDGFA* were defined using GeneArt Strings (TM) and a DNA Fragment Construction system (Thermo Fisher Scientific, Waltham, MA, USA) and were cloned downstream of the luciferase-reporter in the pMIR-REPORT vector (Life Technologies, AM5795M). To measure luciferase activity, phoenix cells (ATCC, CRL-3213) were plated at 80% confluence in 96-well black plates (Corning, NY, USA). After 24 h, cells were transfected with 50 ng of pMIR-REPORT constructs containing the luc-3′-UTR sequences in combination with a Renilla normalization control and 50nM artificial miRIDIAN hsa-miR135b Mimic (Dharmacon; Thermo Fisher Scientific, Waltham, MA, USA) or a miRIDIAN Mimic negative control (Dharmacon; Thermo Fisher Scientific, Waltham, MA, USA, NC-001000-01-05) using the TransIT-X2 transfection reagent (Mirus, MIR6000, Madison, WI, USA). After 24 h, relative luciferase units (RLU) were measured using the Dual-Glo Luciferase Assay System (Promega, Madison, WI, USA) on a GloMax^®^-Multi+Microplate Multimode Reader (Promega, Madison, WI, USA).

### 2.5. Molecular Characterization of PC3 Bone-Metastatic Subclones

#### 2.5.1. RNA and miRNA Isolation

Total RNA including miRNAs was extracted from PCa cells using the miRNeasy Mini Kit (Qiagen, Madrid, Spain) following the instructions provided by the manufacturer. RNA quality was confirmed using the Agilent RNA 6000 Nano Kit in Bioanalyzer 2100 (Agilent Technologies, Santa Clara, CA, USA) and quantified using Nanodrop 1000 Spectrophotometer (Thermo Fisher Scientific, Waltham, MA, USA).

Total RNA including miRNAs was extracted from FFPE human tissue samples using the miRNeasy FFPE kit (Qiagen, Madrid, Spain) following the instructions provided by the manufacturer. Relative miRNA expression was performed by RT-qPCR as mentioned below (see Section 2.5.4 Gene and miRNA expression). A pre-amplification step was added after cDNA obtention: 2.5 µL of diluted cDNA (1:8 in 0.1 × TE) was mixed with 22.5 µL of a mix containing TaqMan PreAmp Master Mix (Thermo Fisher Scientific, Waltham, MA, USA) and a pool of selected TaqMan miRNA assays (Thermo Fisher Scientific, Waltham, MA, USA). Reaction conditions were 95 °C for 10 min, 55 °C for 2 min, and 72 °C for 2 min, followed by 12 cycles of 95 °C for 15 s and 60 °C for 4 min, and finally 99.9 °C for 10 min.

#### 2.5.2. MicroRNA Profiling Analysis

Total RNA was reverse transcribed using an ABI TaqMan miRNA reverse transcription kit (Thermo Fisher Scientific, Waltham, MA, USA). MiRNA expression was profiled using the TaqMan^®^ Array Human MicroRNA A and B Cards Set v3.0 (Applied Biosystems, Waltham, MA, USA). Each card contains 377 human miRNAs, three endogenous small RNA controls (RNU6, RNU44, and RNU48), and a negative control (Ath-miR159a). All reactions were performed as specified in the manufacturer’s protocols. Cards were run in a 7900HT Fast Real-Time PCR System (Applied Biosystems) under the following thermal cycler conditions: 55 °C for 2 min and 95 °C for 10 min, followed by 40 cycles of 95 °C for 15 s and 55 °C for 1 min. Relative expression levels were calculated according to the 2^-ΔΔCt method normalized by RNU48 snRNA. To compare expression levels between groups, the Student’s *t* test was used. The cut-off for statistical significance in the miRNA analysis was a *p*-value ≤ 0.05.

#### 2.5.3. Transcriptomic Array

For gene expression microarray analysis, RNAs were amplified, labelled, and hybridized to an Affymetrix GeneChip^®^ Human Gene 1.0 ST Array (Affymetrix, High Wycombe, UK). Images were processed using the Microarray Analysis Suite 5.0 (Affymetrix). Data from CEL files were normalized and processed using the robust multichip-averaged (RMA) algorithm. The analysis to select differentially expressed genes was based on adjusting a linear model with empirical Bayes moderation of the variance (*p*-value). The selection of differentially expressed genes was based on adjusted *p*-values (false discovery rate (FDR)) <0.001 and absolute LogFC value > 2. MiRNA and transcriptomic data have been deposited in the Gene Expression Omnibus database (Accession number GSE86917).

#### 2.5.4. Gene and miRNA Expression

The reverse transcription of miRNAs was conducted using the TaqMan miRNA reverse transcription kit (Applied Biosystems). For gene expression, 3 μg of RNA was reversed transcribed to first-strand cDNA using the High-Capacity cDNA Archive Kit (Applied Biosystems). Commercially available TaqMan hydrolysis probes (Applied Biosystems) and SYBR Green I were used together with paired primers (KiCqStart^®^ SYBR^®^ Green Primers; Sigma) to analyze the expression of miRNA and gene candidates (listed in the Appendix A), respectively, in an ABI Prism 7900HT qPCR system (Applied Biosystems). Relative miRNA and gene expression levels were calculated according to the 2^-ΔΔCt method normalized using RNU48 snRNA and the average between GAPDH and HMBS, respectively.

#### 2.5.5. miRNA Expression in Plasma

Total RNA was extracted from 500 µL of plasma using a Maxwell RSC miRNA Plasma and Serum Kit (Promega, Madison, WI, USA) following the manufacturer’s instructions and eluted in 40 µL of RNase free water. All samples were spiked with 25 fmol/μL of Caenorhabditis elegans miR-39 for use as a normalizer in downstream analyses.

miRNA expression levels were evaluated using individual Taqman Advanced miRNA assays (Thermo Fisher Scientific, Waltham, MA, USA, #A25576): hsa-miR-135b (ID 478581_mir) and cel-miR-39 (ID 478293_mir).

According to the manufacturer’s instructions, 10 ng of total RNA was first poly-A tailed and, after adaptor ligation and reverse transcription, pre-amplified for 20 cycles. The pre-amplification product was subsequently diluted 1/5 in 0.1% TE buffer, pH 8.0. A qPCR test was performed using a Taqman Fast Advanced Master mix on a QuantStudio 7 Pro realtime PCR device (Thermo Fisher Scientific, Waltham, MA, USA). Relative expression levels were calculated according to the 2^-ΔΔCt method. To compare expression levels between groups, the nonparametric Mann–Whitney U test was used.

### 2.6. Bioinformatics Analysis

#### 2.6.1. Pathway Analysis

Gene pathway analysis was carried out by the Anaxomics Biotech Company, considering gene expression data from the transcriptomics array and ranking genes according to their statistical significance (LogFC). The enrichment was run over several sets including GO terms (Biological Process, Cellular Component, Molecular Function) according to EMBL-EBI/UniProt-GO [43], KEGG pathways [44], pathological conditions, and motives included in de BED (Anaxomics property database) [45], PharmaGKB pathways [46], the pathways from Small Molecules Pathway database (SMPDB) [47], and the transcription factor profiles included in the TRUST database [48]. Only the sets that showed statistically significant enrichment after multi-testing using the Benjamini–Hochberg correction are presented (FDR < 0.05).

#### 2.6.2. Analysis of miRNA Targets

In silico prediction of potentially miR-135b target genes was performed using a miRWalk platform by four independent algorithms (TargetScan [49], miRanda [50], miRWalk [51], and RNAhybrid [52]). Only targets that were predicted by four algorithms were considered for subsequent studies. KEGG pathways and GO (biological process (BP)) gene sets were also downloaded from miRWalk platform and used to analyze the biological significance of miR-135b predicted targets. A *p*-value ≤ 0.05 was considered statistically significant.

#### 2.6.3. Analysis of PCa Genomic and Transcriptomic Datasets

The TCGA dataset repositories and datasets from different studies with PCa patients were interrogated using the cBioportal website [53]. The clinical information regarding overall and disease-free survival was considered to be correlated with the expression levels of miR-135b target genes. Kaplan–Meier Pots and Logrank Test *p*-values were obtained.

## 3. Results

### 3.1. In Vivo Selection for Bone-Metastasizing PC3 Cells

Tumors contain heterogeneous populations with different abilities to metastasize to distant organs such as bones. In order to isolate highly metastatic subpopulations that attach preferably to bone, we generated a bone-seeking variant of the PC3 cell line using in vivo passaging in immunodeficient mice. To this end, PC3 luciferase-expressing cells were inoculated intracardiacally (i.c.). Mice developed bone lesions two weeks after cell inoculation and animals were monitored using bioluminescence (BLI) imaging until the end of the study. After the sacrifice, we isolated PC3-derived metastasis cells from the long bones (femur and tibia), which were shortly expanded in cultures and injected into a new group of mice (Figure 1A,B). The metastatic subclone named PC3-BM, isolated from the third in vivo round, presented a higher incidence of metastasis to long bones compared with the parental PC3 cell line (PC3-P) (i.e., 100% versus 20%) (Figure 1C). Moreover, the distribution and number of bone metastases generated by PC3-BM cells was higher in leg, spinal cord, and scapula areas compared with other bones (Figure 1D). As expected, PC3 cells growing into the bones caused osteolytic lesions as analyzed using micro-CT imaging and 3D modelling (Figure 1E). These results suggest that PC3-BM cells display higher efficiency in nesting to the bone and growing into the bone environment.

### 3.2. Transcriptional Analysis of PC3-BM Cells Revealed Multiple Bone Metastatic-Related Genes

To describe the molecular changes associated with cells’ ability to migrate to the bone tissue, we performed a whole-genome expression microarray analysis comparing PC3-P and PC3-BM cells. Principal component analysis segregated both cell lines, which indicated a different transcriptional pattern (Figure 2A). This comparison yielded a list of 95 differentially expressed genes with an absolute log FC value ≥ 2 and an adjusted *p*-value (FDR) < 0.001 (Figure 2B). Among them, 61 genes were found to be upregulated in PC-3-BM cells, whereas 34 were downregulated (Appendix A).

To better understand the role of those deregulated genes in bone metastasis development, we performed a KEGG pathway analysis among other analyses (see Section 2) (Appendix A). The differentially expressed genes were grouped into different up and down cancer-related general processes (Figure 2C,D). We observed enrichment in functions related to tumor aggressiveness, such as cancer metastasis, cell division, immune system, bone-related process, and angiogenesis in PC3-BM cells. Heatmaps depict the significantly deregulated genes in the most relevant process related to bone metastasis (Figure 2E). Thus, the gene expression analysis of the PC3-BM cells compared with the parental PC3 cell line revealed the presence of distinct transcriptional patterns related to the acquired bone tropism of these cells.

### 3.3. Differential miRNA Expression Profile of In Vivo Selected Metastatic Prostate Cancer Cells

To identify the differentially expressed miRNA profiles in PCa cells that could mediate tumor progression within the bone microenvironment, we compared the miRNA profiles of PC3-BM and PC3-P cells using TaqMan Array MicroFluidic Cards. We identified 16 differentially expressed miRNAs with a FC >1.5 and a *p*-value ≤ 0.05; 11 miRNAs were downregulated whereas 5 were upregulated in the bone metastatic cell line compared with the parental one (Figure 3A,B). Microarray results were further validated by RT-qPCR and confirmed that miR-135b, miR-200b, miR-18a, miR-425, miR-194, and miR-19a were downregulated in PC3-BM cells (Figure 3C).

We then analyzed the expression of the selected miRNAs in a cohort of 42 primary formalin-fixed paraffin-embedded (FFPE) PCa samples from radical prostatectomies selected by the Pathology Department of Vall Hebron University Hospital (Appendix A). Patients were followed up for 7–10 years after surgery and divided into two groups: with or without biochemical recurrence (BCR). MiR-135b, miR-194, and miR-19a presented significantly lower expressions in patients with worse prognoses (BCR) (Figure 3D, *p*-value < 0.05). Following to the results obtained in the cell line model and patient samples, we chose miR-135b, miR-200b, and miR-19a for further functional studies.

### 3.4. Overexpression of miR-135b, miR-200b, and miR-19a Decreases Migration of PC3-BM Cells

To study the role of the selected miRNAs in promoting the growth of tumor cells in the bone, we first co-cultured PC3-BM cells with osteoblasts cells (hFOB) or their conditioned media and tested whether the bone microenvironment had an impact on their migration capacity. The presence of osteoblasts or their conditioned media significantly increases the migratory capacity of PC3-BM cells (Figure 4A). This effect was osteoblast-specific, since the conditioned media of two other non-tumoral cell lines (i.e., HEK293 and Het1A) did not produce the same effect (Figure 4A).

Next, we further analyzed the impact of overexpressing miR-135b, miR-200b, and miR-19a on the migratory capacity of PC3-BM cells using a transwell migration assay. When miR-135b and miR-19a were overexpressed in PC3-BM cells in the presence of osteoblasts, we observed a reduction in the migratory capacity of PC3-BM cells compared with miRNA-control-transfected cells (Figure 4C, *p*-value < 0.001). The overexpression of miR-200b showed a similar trend but did not reach statistical significance. To rule out the conclusion that the observed effect on migration was not due to impaired proliferation, a cell proliferation assay was performed under the same experimental design. As observed in Figure 4D, miRNA overexpression decreased PC3-BM viability by less than 10% compared to the negative control (NC), ensuring that the observed reduction of cell migration was not due to a reduction in the cell number.

Taken together, these results indicate that miR-135b and miR-19a downregulation observed in bone metastatic clones may be involved in the development or progression of bone metastasis of PCa.

### 3.5. Integration Data Analysis Identifies Biomarkers of Metastasis and Pathways Related to PCa Bone Metastases

Dysregulation of miR-135b levels has been associated with cancer invasion and the migratory abilities of multiple cancer types [28,34,37,54]; however, its role and the molecular mechanism of its target genes in the regulation of bone metastasis of PCa remain to be elucidated. The fact that miR-135b showed no effects on proliferation, coupled with these observations, led us to focus our next studies on this particular miRNA.

Our findings on reduced miR-135b levels in the highly bone-metastatic cell line prompted us to analyze the landscape of molecular pathways to which this non-coding RNA could be related, concerning PCa pathogenesis.

To this end, a whole transcriptome 3′UTR search for miR-135b binding sites using the miRWalk2.0 platform [51], followed by a pathway analysis of the predicted miR-135b target genes (Appendix A), was conducted. Relevant molecular pathways for the bone metastasis phenotype are summarized in Figure 5A; they are represented by the number of genes in each set with a *p*-value ≤ 0.01. Among them, functions related to cancer, and especially to disseminative and invasive processes such as cell–cell or cell–matrix adhesion, cell motility, and cytokine-mediated signaling pathways, were enriched. In accordance with our model, genes related directly to the bone, such as osteoclast differentiation, calcium signaling pathway and mineral absorption, were overrepresented some of them obtained by both sources, i.e., KEGG and GO term (Biological Function) (Figure 5A).

To further investigate molecular mechanisms under miR-135b regulation, we screened the overlap between the putative miR-135b target genes, obtained from four different algorithms (TargetScan, miRWalk, RNAhybrid, and miRanda) (Appendix A), and our transcriptomic array data (Appendix A). Interestingly, miR-135b target genes *ANGPT2*, *JAKMIP2*, *PLAG1*, *GRID2*, *VTI1B*, *PDGFA*, and *CYBRD1* were also found to be significantly upregulated in PC3-BM compared with PC3-P cells (Figure 5B). Consequently, the relative expression levels of these targets were technically validated by RT-qPCR, confirming that all of them presented significantly higher levels in the bone metastatic clone (Figure 5C). However, *GRID2* and *CYBRD1* were not considered for later studies due to their low expression levels.

### 3.6. miR-135b Regulates JAKMIP2, PLAG1, PDGFA, and VTI1 Levels

To further confirm the direct relationship between miR-135b and its putative differentially expressed target genes, the levels of miR-135b were modulated by transient transfection of miRIDIAN Mimics. Upon miR-135b overexpression *PLAG1*, *JAKMIP2*, *PDGFA*, and *VTI1b* levels were significantly decreased in PC3-BM cells compared with control transfected cells (Figure 5D). Furthermore, luciferase-reporter assays carried out after miR-135b overexpression showed a significant reduction in luciferase activity for all 3′UTR-engineered regions, indicating a direct modulation of these genes by miR-135b (Figure 5E). Together, these results suggest that the altered expression of *PLAG1*, *JAKMIP2*, *PDGFA*, and *VTI1b* is due, at least in part, to the reduction of miR-135b expression levels in bone metastasis.

### 3.7. miR-135b and Its Targets, JAKMIP2, PLAG1, and PDGFA, Correlate with Poor Prognosis of PCa Patients

As miR-135b expression levels were significantly lower in FFPE samples from patients with biochemical recurrence (Figure 3D), we wanted to further evaluate the miR-135b levels as a potential predictor of aggressiveness in PCa human samples. To test this, 40 plasma samples from PCa patients, including localized (*n* = 11), locally advanced (*n* = 9), and metastatic disease (*n* = 20) and 9 healthy controls, were selected to analyze circulating levels of miR-135b. As shown in Figure 6A, circulating levels of miR-135b were significantly lower in blood samples from patients with locally advanced and metastatic disease compared with samples from patients with localized disease. In particular, miR-135b circulating levels were lower in T3-4N0M0 and T1-4N0-1M1(Figure 6A). Furthermore, when miR-135b levels were studied based on their Gleason scores, their correlation with advanced PCa was once again observed: miR-135b expression levels were significantly lower in patients with higher Gleason scores (Figure 6B).

Finally, data mining of publicly available PCa-gene-expression datasets (i.e., TCGA repository) was conducted to find out whether the identified miR-135b target genes correlated with clinical variables in PCa. The combined expression of the four candidate genes (*PLAG1*, *JAKMIP2*, *PDGFA*, and *VTI1b*) correlated with disease-free time (*p*-value = 0.00002, 95% CI) (Figure 6C). Patients with abnormal levels of those four genes (*n* = 64) showed a significant decrease in disease-free time when compared with patients with unaltered levels (*n* = 1267). Individual analyses of the Kaplan–Meier curves for each gene (Appendix A) revealed that this prognostic value was mainly associated with high levels of *PLAG1*, *JAKMIP2*, and *PDGFA*. Similar results were obtained when overall survival was evaluated, with a significant shortening (20%) of the 10-year survival rate in the group of patients with altered levels of the four studied genes (*n* = 110) compared with the unaltered group (*n* = 1366) (*p*-value = 0.003, 95% CI) (Figure 6D). In this case, the effect seemed to be related to high levels of *PLAG1* and *PDGFA* (Appendix A). Our results support the clinical relevance of the identified miR-135b and their target genes in bone metastasis patients of PCa.

## 4. Discussion

Although the survival of PCa patients has improved over the last two decades, once the disease disseminates to distant organs such as bone, the prognosis drastically worsens, with only 30% of patients surviving more than 5 years [2]. Bone metastasis represents a scenario that is difficult to treat due to pain, fracture risk, and decreased quality of life. It is increasingly appreciated that the microenvironment is crucial in supporting metastases formation; disseminated tumoral cells must acquire the capability to reprogram in their new microenvironment to support their growth and enhance the formation of clinically relevant metastases [55]. Thus, bone-specific metastasis likely results from both tumor-cell-intrinsic mechanisms and the microenvironment provided by the bone. Having in mind the “seed and soil” hypothesis described by Paget, which explained why certain cancers favored developing metastasis in specific organs [56,57], understanding the distinct organ-specific mechanisms that enable metastatic growth is of crucial importance. Our study reveals novel molecules involved in osteotropism and/or in creating a permissive pre-metastatic niche for PCa cells. A better understanding of these mechanisms would lead to the development of more specific therapeutic strategies to hinder or prevent the establishment of bone metastatic lesions and consequently improve clinical outcomes for patients with PCa.

Animal models of skeletal metastasis are essential to study the molecular pathways of cancer spread. In this study, we used a preclinical model based on intracardiac injection of PCa cells into nude mice to faithfully reproduce the dissemination steps of bone metastasis [58]. Cancer cells established in the bone niche showed an increased incidence of bone lesions (up to 100%) after three rounds of in vivo selection, compared to a 20% incidence of cells prior to in vivo passaging. Interestingly, the PC3-BM cells metastasize with high penetrance to long bones, spine, and ribs, as has been described in metastatic PCa patients [5].

In vivo results suggest that the parental PC3 cells were reprogrammed as a consequence of the interaction with the bone microenvironment, which underpins the ability of PC3-BM to disrupt bone homeostasis, instigate bone degradation, and form a more hospitable metastatic niche. We therefore took advantage of both cell populations to elucidate the mechanisms enabling the capacity to metastasize to the bone. By carrying out a comparative whole-genome gene expression analysis, we revealed an upregulation of a set of genes related to cancer metastasis, angiogenesis, the extracellular matrix, and bone-related processes. Looking into detail in each of these categories, genes previously related to bone tropism were found to be overexpressed in PC3-BM cells, such as *CXCR4* and its ligand *CSF1*, as well as *CCL2*, all of them directly involved in signaling pathways regulating circulating tumor cells (CTCs) homing to bone [59,60]. Moreover, *ANGPT2* and *PDGF* are also relevant molecules for cancer-cell-survival in the bone marrow niche for ER+ breast and PCa cells, respectively [61,62]. The acquired gene expression profile of PC3-BM cells notarized the reliability of our preclinical model and could explain, at least in part, the distinct efficacy of current treatments on disease progression due to the intratumor heterogeneity.

Giving the relevance of miRNAs as key post-transcriptional gene repressors in multiple steps of cancer progression, including those osteotropic cancers [38], we further extended the study to evaluate changes to these molecules. As these small non-coding RNAs are able to regulate several targets simultaneously, we consider them to be promising future therapeutic candidates to target advanced PCa disease. Using high-throughput analysis, we obtained a set of top differentially expressed miRNAs when the PC3-BM profile was compared with the parental cell line. To validate our findings, we performed a bibliographic review to study our dataset of miRNA candidates (*p*-value ≤ 0.05) (Appendix A). All miRNAs had previously been related to cancer and metastasis; 50% of them were described in PCa metastasis and specifically miR-200b, miR-34a, miR-194, or miR-210, among others, had been previously related to the bone metastasis process, thus indicating that our model was representative of the disease [25,37,63,64,65,66,67,68,69,70,71,72,73,74,75,76,77,78,79,80,81,82,83].

Furthermore, we explored the association between those miRNAs significantly downregulated in the PC3-BM subclone (miR-135b, miR-422a, miR-200b, miR-18a, miR-425, miR-194, and miR-19a) and the prognosis of PCa in a cohort of primary tumors. Interestingly, those patients that had developed BCR (worse prognosis) also presented significant less expression of miR-135b, miR-194, and miR-19a compared with patients without BCR. This indicates that these miRNAs could bear a potential prognostic value. In particular, miR-135b had previously been associated with both an oncogenic and a tumor-suppressor role in a context-dependent manner [84,85]. Wang et al. [37] analyzed the expression of miR-135b in primary PCa samples and nonmalignant samples using RT-qPCR; their findings revealed that miR-135b was significantly down-regulated in PCa tissues compared with noncancerous tissues. Their results also indicated that the expression levels of miR-135b correlated with the pathological T stages of PCa and demonstrated significant inverse correlations between the expression of miR-135b and the levels of total and free PSA. In our work, the circulating levels of miR-135b have been determined for the first time, and the results indicate, once again, that this miRNA has potential as a biomarker and, after appropriate validation in a larger cohort of patients, it could potentially be used in clinical practice as a tool to assess patients’ prognoses.

On the other hand, miR-135b has been identified as a direct regulator of androgen receptor (AR)-protein-levels in PCa [34,35,36]. Although, in our model based on PC3, the cell line that presents undetectable levels of AR, the effect of miR-135b on the selection of bone as a metastatic niche would most likely not be mediated by AR. It is important to mention that further studies should be conducted to elucidate the potential role of miR-135b in the acquisition of PCa which is resistant to androgen deprivation therapy.

Previous studies have reported that miR-135b may contribute to cell migration and invasion in PCa cells; however, its role in the development of bone metastasis has not been documented thus far. To this end, in vitro functional assays analyzing miR-135b’s role in migration using an osteoblast cell line were carried out. It is known that prostate tumor cells enter in touch with bone lining cells when located in the bone marrow. Bone lining cells are quiescent or premature osteoblasts; and therefore, the immature human hFOB cell line is a particularly good model for co-culture studies using PCa cells [86].

Our results demonstrate that PC3-BM cancer cells had a significantly increased migration capacity when they were co-cultured with the osteoblast cell line compared with the parental cells. This phenotype reinforces our in vivo model of bone homing and the colonization potential of PC3-BM cells. Interestingly, this effect was reversed when miR-135b was overexpressed, which suggests that this miRNA could be involved, at least in part, in the bone-homing process. This result evokes, for the first time, the possible implication of miR-135b in the bone metastasis process of PCa.

Furthermore, the released factors that mediate the crosstalk between the primary tumor cells and those from metastatic niche are crucial [87], providing the exosomes with a relevant role in this intercellular communication [88]. Recently, the signature of aberrantly expressed miRNAs has been identified in PCa- tumor-derived exosomes. Among them, miR-135b was found infra-expressed in hypoxia-related biomarkers in PCa [63]. Since it has been shown how exosomes are directed to the specific metastatic organs enabling organotropic metastatic growth [88,89], the deregulated expression of miR-135b in this intercellular communication system could partly explain its role in bone tropism.

To unveil the landscape of molecular pathways in which miR-135b target genes are involved we extended our study to analyze the differentially expressed genes in the more metastatic cell line. miR-135b-predicted targets were enriched in oncogenic signaling pathways, metastasis, and bone remodeling-related genes in PC3-BM cells. After the overlapping of miR-135b-predicted targets with our transcriptomic data, *VIT1b*, *JAKMIP2*, *PLAG1*, and *PDGFA* were validated as direct targets of miR-135b. Additionally, data mining of PCa gene expression analysis showed that patients who presented higher levels of the four candidate genes (*PLAG1*, *JAKMIP2*, *PDGFA*, and *VTI1b*) had worse overall survival compared with those patients with unaltered levels. In concordance with that, high levels of *PDGFA* have previously been related to tumorigenesis, progression of PCa, and the biology of bone tissue. Moreover, *PDGFA* has long been associated with poor prognosis and metastasis [90], cell proliferation, and migration [91], and it has also been described as an osteogenesis-associated gene in PCa progression [92,93]. Otherwise, to date, it has not been reported the involvement of *PLAG1*, *JAKMIP2*, and *VTI1b* in PCa disease. However, *PLAG1* is a zinc-finger transcription factor that has been described as an oncogene in other type of tumors. Previous studies demonstrated that it could promote anoikis resistance and tumor metastasis in lung cancer [94], and its silencing also promoted cell-chemosensitivity in ovarian cancer [95]. The oncogenic capability of *PLAG1* seemed to be mediated, at least partly, by the IGF-II-mitogenic signaling pathway [95,96]. Interestingly, *JAKMIP2* (Janus Kinase and Microtubule Interacting Protein 2) and *VTI1b* (Vesicle Transport Through Interaction With T-SNAREs 1B) have been linked to vesicle transport. *JAKMIP2* was reported to be a component of Golgi matrix and acting as a Golgi protein by negatively regulating transit of secretory cargo [97]. On the other hand, *VTI1b* mediates vesicle transport pathways and vesicle trafficking. For instance, it has been shown that the levels of *VTI1b* accommodate the volume and timing of post-Golgi cytokine trafficking in macrophages [98,99]. Since it has been well established that circulating tumor cells are selectively and actively modified by the primary tumor before metastatic spread has even occurred [100], the vesicle-related function of both *JAKMIP2* and *VTI1b* could be relevant for the tumor cells localized at the primary site, which need to prepare the future-tissue colonization through the establishment of the pre-metastatic niche.

In summary, our results show that PCa cells that metastasize to bone possess features that enable them to efficiently utilize the bone microenvironment, creating a permissive pre-metastatic niche for PCa cells. Therapeutic strategies based on miRNA modulation are shown to be promising approaches for treatment of heterogeneous disease such as cancer [40]. In this sense, a comprehensive group of miRNAs associated with bone homing and tumor aggressiveness were recovered in the osteotropic tumor subclone supporting the robustness of our metastatic in vivo model of PCa. Specifically, our results suggest that the deregulation of miR-135b could be essential for bone selection as a metastatic niche for PCa, and its targets *PLAG1*, *JAKMIP2*, *PDGFA*, and *VTI1b* could also be mediating the establishment of bone metastasis. miR-135b and its target genes could potentially be considered as predictive biomarkers of bone dissemination, having a determining impact on clinical decisions and therefore improving PCa patients’ outcomes.

## 5. Conclusions

The clinical management of PCa patients with bone metastases remains a challenge due to the inability to obtain long-term benefits post treatment and the development of resistance to current standard therapies. The process of metastasis requires multiple steps which include local tumor invasion, intravasation, survival in the bloodstream, extravasation, and the colonization of the distal sites, which leads us to think that, to deal with this phenomenon, it would be convenient to design a therapeutic strategy that encompasses the different factors involved. We suggest that the modulation of miRNAs achieved at the primary tumor may be a promising approach to prevent skeletal metastasis. Specifically, our findings reveal that miR-135b could play an important role in the selection of bone as a metastatic niche, and this effect could be mediated by its target genes, *PLAG1*, *JAKMIP2*, *PDGFA*, and *VTI1b*. Furthermore, miR-135b showed potential as a poor prognostic biomarker.

## Figures and Tables

**Figure 1 cancers-13-06202-f001:**
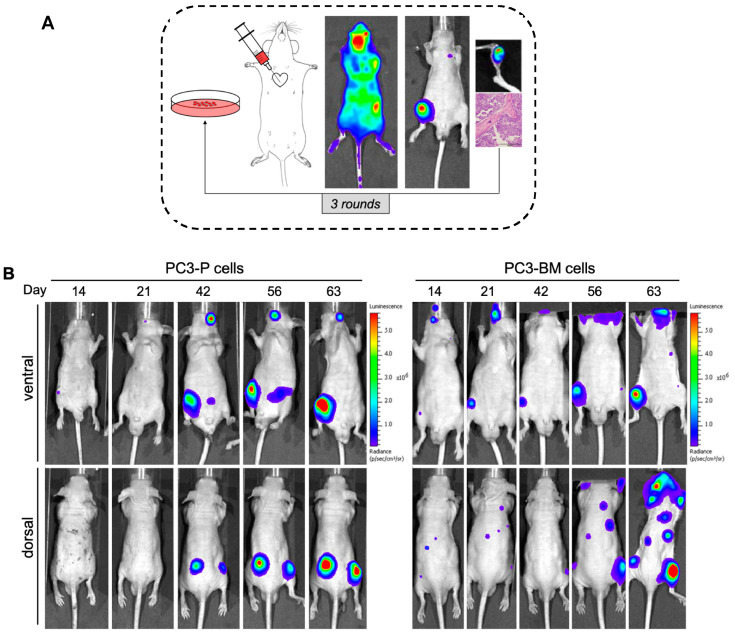
Selection of human PC3 prostate tumor cells with increased bone metastatic phenotype. (**A**) View of the sequential method for the in vivo selection of PCa cells with bone tropism. (**B**) Representative bioluminescence imaging (BLI) of the metastatic distribution pattern of parental and PC3-BM cells over time. (**C**) Incidence of bone metastases for PC3 cells in long bones after the indicated rounds of in vivo selection. Results are expressed as the percentages of metastases in long bones per total number of long bones in analyzed mice. (**D**) Distribution and incidence of metastases in the different isolated bones. Results are expressed as the percentages of metastases detected by BLI imaging per total number of animals injected in each round. (**E**) Representative micro-CT images and 3D modelling showing osteolytic lesions caused by PCa cells. Black and yellow arrows show the metastatic lesions.

**Figure 2 cancers-13-06202-f002:**
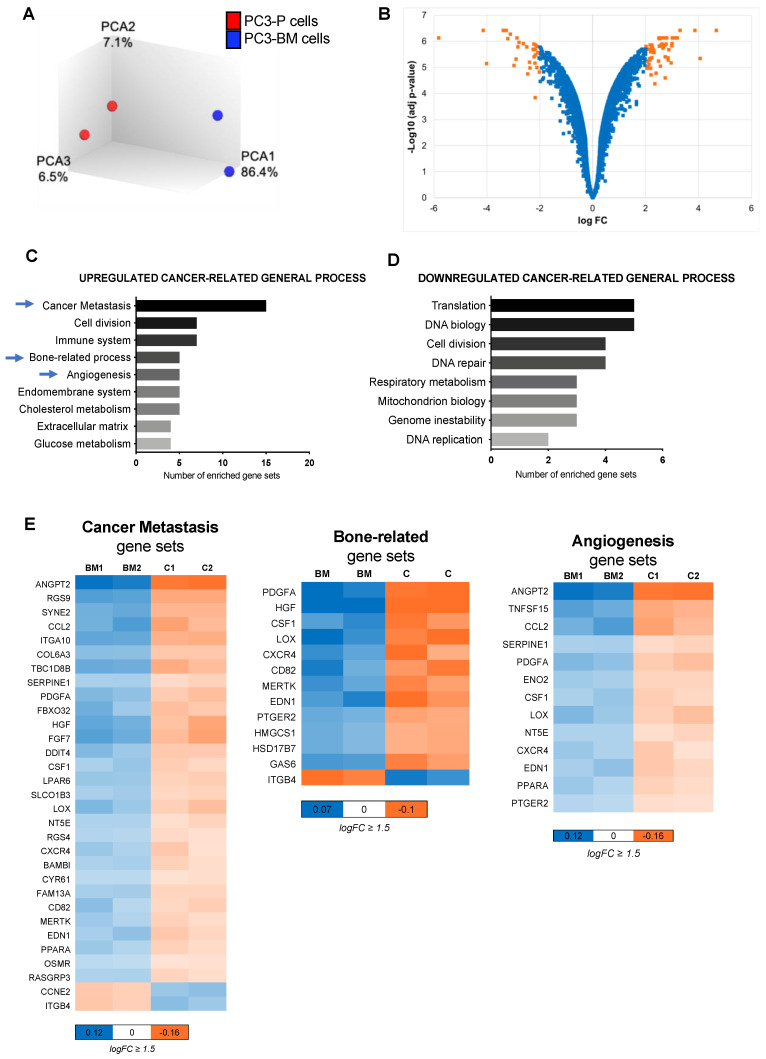
Gene expression analysis of the bone metastatic subclone. (**A**) Principal component analysis illustrates segregation of distinct expression profiles of bone metastatic subclone PC3-BM and PC3-P cell lines. (**B**) Volcano plot of differential expression analysis of PC3-BM compared with PC3-P cell line. (**C**,**D**) Analysis of cancer-related gene sets using the transcriptome data. Graphs represent the total number of statistically significant up-and down-regulated enriched gene sets after multi-testing Benjamini–Hochberg correction, with FDR < 0.05, assigned to different cancer-related general process. (**E**) Heat maps indicating the genes included in the most relevant enriched gene sets for bone metastasis development. The color key shows relative expression levels of the differentially expressed genes, BM: PC3-BM and C: PC3-P cells.

**Figure 3 cancers-13-06202-f003:**
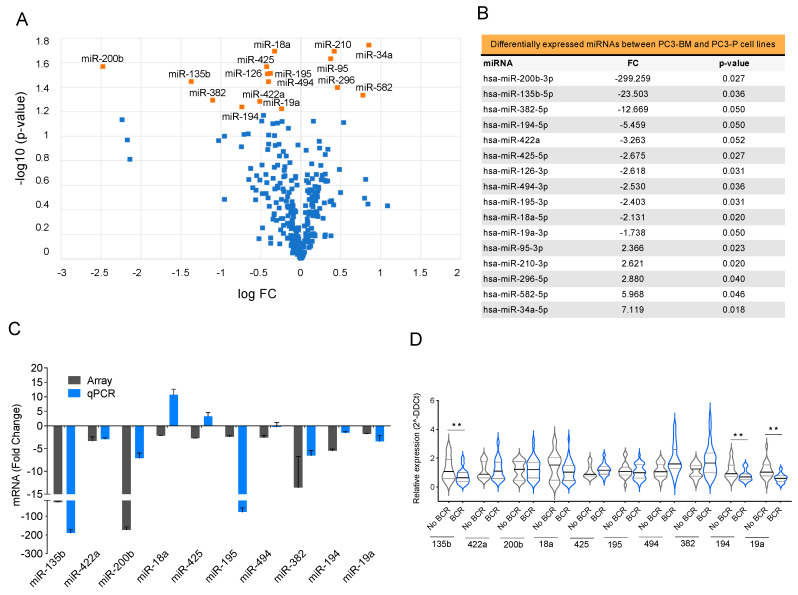
Selection of miRNAs with potential roles in bone metastasis (**A**) Volcano plot of miRNA profiling analysis of PC3-BM compared with PC3-P cell line. (**B**) Differentially expressed miRNAs between PC3-BM and PC3-P cell line. The table represents the statistically significant deregulated microRNAs by t-Student analysis with *p*-value ≤ 0.05 and FC > 1.5. (**C**) RT-qPCR analysis validation of the microarray results. (**D**) Relative miRNA expression levels in primary tumors from PCa patients with and without biochemical recurrence (BCR) after radical prostatectomy. RNU48 was used as an endogenous control for all miRNA expression analyses. Values that are significantly different by *t*-test analysis are indicated by ** < 0.01.

**Figure 4 cancers-13-06202-f004:**
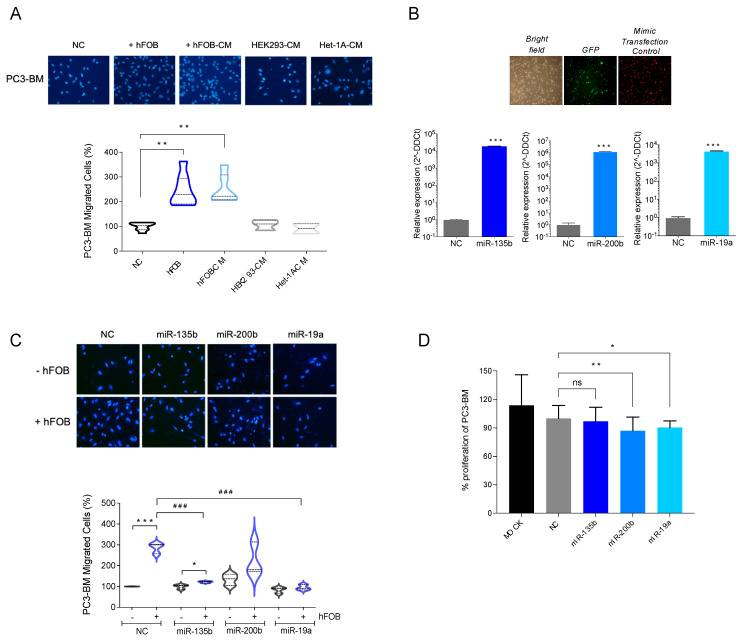
Functional study of selected miRNAs. (**A**) Boyden chamber migration assay. hFOB: human osteoblast cell line. CM: conditioned media. (**B**) Relative miRNA expressions of miR-135b, miR-200b and miR-19a in transfected cells compared to cells transfected with mimic. Confirmed to have minimal sequence identity with miRNAs in human (negative control: NC). Images are representative of PC3-BM cells transduced with the firefly luciferase gene-coding region cloned upstream of the green fluorescent protein (GFP). Red fluorescence was used to monitor miRNA transfection efficiency of Dy547-labelled miRIDIAN microRNA Mimic Transfection Control. (**C**) Boyden chamber migration assays with PC3-MB transfected cells with NC or with miR-135b, miR-200, and miR-19a microRNA mimics and stablishing co-culture system with hFOB. (**D**) Proliferation assay of PC3-BM non-transfected (MOCK) or after NC, miR-135b, miR-200, and miR-19a transfection at 96 h (*n* = 6/condition). All bar graphs and plots represent the mean ± SEM of three independent experiments. Values that are significantly different by *t*-test analysis are indicated by * < 0.05, ** < 0.01, and *** < 0.001 or ^###^ < 0.001, ns: no statistically significant. P: PC3 cells; BM: PC3-BM cells; hFOB: osteoblast cell line; CM: conditioned media.

**Figure 5 cancers-13-06202-f005:**
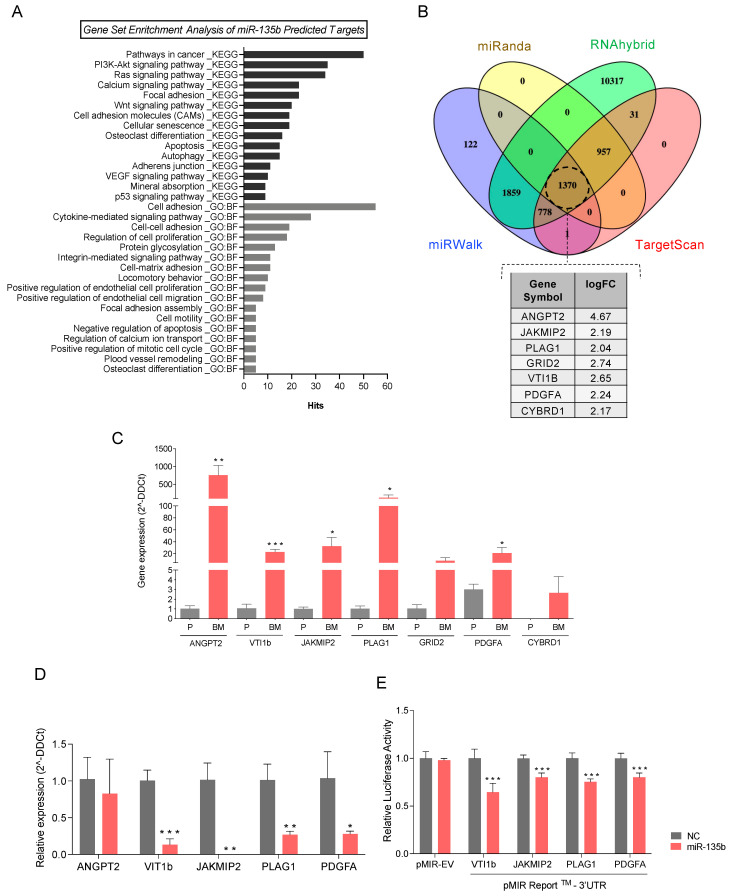
MiRNA-135b target genes involved in bone metastasis. (**A**) Cancer-and metastasis-related enriched gene sets with *p*-value ≤ 0.01 from miRNA-135b predicted-target genes. KEGG pathways and GO term (biological function: BF) gene sets were downloaded from the miRWalk platform and used for the pathways analysis. (**B**) Venn diagram illustrating the overlap among the miR-135b-predicted target genes of four different algorithms (TargetScan, miRanda, miRWalk and RNAhybrid) using miRWalk platform. The table shows those target genes significantly over-expressed in the transcriptomic data of PC3-BM compared with PC3-P found in common with the predicted target gene set. (**C**) RT-qPCR analyses of genes that are predicted targets of miR-135b in PC3-BM compared with PC3-P. (**D**) Relative mRNA expression levels of the indicated putative miR-135b target genes at 48h post-transfection with miR-135b mimic and miR-Control (NC) in PC3-B©ells. (**E**) Luciferase assay performed in phoenix cells co-transfected with the indicated 3′UTR luciferase-reporter vectors and miR-135b mimic or miR-Control (NC). Graph represents luciferase activity normalized to the renilla internal control (*n* = 5). All bar graphs represent the mean ± SEM. Values that are significantly different by *t*-test analysis from NC group are indicated by * < 0.05, ** < 0.01, and *** < 0.001.

**Figure 6 cancers-13-06202-f006:**
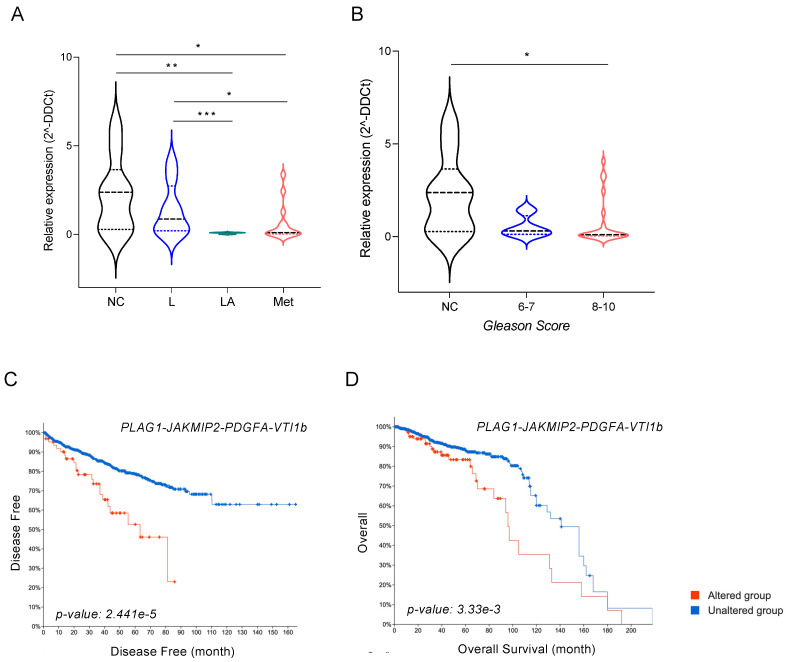
miR-135b and its targets *JAKMIP2*, *PLAG1*, and *PDGFA* as prognostic biomarkers. (**A**) Relative expression levels of miR-135b in plasma from healthy controls (NC) and patients with localized PCa (L; T1-2N0M0), locally advanced PCa (LA; T3-4N0M0), and metastatic PCa (M; T1-4N0-1M1). (**B**) Relative expression levels of miR-135b in plasma from healthy controls (NC) and patients with different Gleason scores. (**C**) Correlation between disease-free time (in months) and (**D**) overall survival with the expression levels of miR-135b-validated targets. Patients’ data were retrieved from different PCa studies and TCGA Repository using the cBioportal Visualization platform. Values that are significantly different by Mann–Whitney U test analysis are indicated by * < 0.05, ** < 0.01, and *** < 0.001.

## Data Availability

MiRNA and transcriptomic data have been deposited in the Gene Expression Omnibus database (Accession number GSE86917).

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
