# Peer review of "Loss of microRNA-135b Enhances Bone Metastasis in Prostate Cancer and Predicts Aggressiveness in Human Prostate Samples"

_cancers, 2021, doi:10.3390/cancers13246202_

Round 1

Reviewer 1 Report

Just concerning GAPDH, I will suggest using also another housekeeping such us HPRT1. GAPDH es high expressed and doing an overall with other housekeepings with different expression patterns will minimize the disperssion of the data.

Reviewer 2 Report

The authors address all the points and the manuscript can now be accepted in its present form.

Author Response

This manuscript is a resubmission of an earlier submission. The following is a list of the peer review reports and author responses from that submission.

Round 1

Reviewer 1 Report

The manuscript by Olivan et al described that miR-135b could play an important role in the selection of bone as a metastatic niche affecting the expression of several target genes such as PLAG1, JAKMIP2, PDGFA and VTI1b. The manuscript is linear, well-written and very interesting for its potential clinical impact. The study design is good and the results well described. the conclusion is supported by the results.

The authors suggested that mir-135b could be a therapeutic target and a prognostic biomarker. I suggest to measure circulating levels of this miRNA in serum samples of prostate cancer patients stratified by Gleason score in order to suggest its potentaila use in clinical practice as tool to assess patient prognosis.

Reviewer 2 Report

Minor comments:

  1. PCa abbreviation is not always used.
  2. Summary figure abstract is recomended to better understanding.

Other points to take into account in different sections:

  1. Introduction section. In my opinion this section should be completed, there is no many details of miRNAs in PCa, and there is extensive literature about it. Moreover , more details of miR-135b in other tumors should be interesting to be added. In my opinion this section is quite short and not completed.
  2. Materials and methods: According to cell lines, why you only use PC3? there are also commercial lines such as LNCap that should be interesting to be proved, to see differences and contrast results.
  3. FFPE samples are too small to prove your results, and you shoul detail how you choose the area of analysis. Moreover, expression analysis are modified by formalin treatment, so proven in not fixed samples should be recomended to asses the utility of the marker.
  4. Could you explain better why you use RNU48 as housekeeing? results could be totally different according the normalization with a different housekeeping.
  5. GADPH is not the best option, and you used in 2.5.4. why?
  6. Result section: there is too many information in Figure 2, this makes it difficult to follow it.
  7. Figure 3 are too small to be understood, and figure 5 A is the same.
  8. Discussion section: More robust data of the strength in the miRNA that you are focusing the discovery should be included in the text.
  9. Final comment. In my opinion you should prove in more samles and in different samples, fresh tissue, blood...to totally  prove your suspicious in this miRNA. 

Reviewer 3 Report

MicroRNA-135b enhances bone metastasis in prostate cancer and predicts aggressiveness in prostate human samples by Olivan et al is an interesting study. However, it needs some more evidences to strengthen the manuscript.

Comment 1. The study is interesting; however, the authors show only metastasis in only one cell line PC3, it is important to show hormone therapy sensitive and hormone sensitive or any other prostate cancer cell line (C4-2 or VCap) to make a commonality.

Comment 2.  Does Androgen receptor (AR) have any link with this bone metastasis and MicroRNA-135b 

Comment 3.  Almost all the experiments are carried out only in PC3 cells and some of the experiments need to be validated in other prostate cancer cells